METHODS

# Improving polygenic prediction from summary data by learning patterns of effect sharing across multiple phenotypes

**Deborah Kunkel**[1]*, **Peter Sørensen**[2], **Vijay Shankar**[3], **Fabio Morgante**[3,4]*

**1** School of Mathematical and Statistical Sciences, Clemson University, Clemson, South Carolina, United States of America, **2** Center for Quantitative Genetics and Genomics, Aarhus University, Aarhus, Denmark, **3** Center for Human Genetics, Clemson University, Greenwood, South Carolina, United States of America, **4** Department of Genetics and Biochemistry, Clemson University, Clemson, South Carolina, United States of America

* dekunke@clemson.edu (DK); fabiom@clemson.edu (FM)

**Data Availability Statement:** The genotype and phenotype data used in our analyses are available from UK Biobank (https://www.ukbiobank.ac.uk/). All code implementing the simulations and data

## Abstract

Polygenic prediction of complex trait phenotypes has become important in human genetics, especially in the context of precision medicine. Recently, *mr.mash*, a flexible and computationally efficient method that models multiple phenotypes jointly and leverages sharing of effects across such phenotypes to improve prediction accuracy, was introduced. However, a drawback of *mr.mash* is that it requires individual-level data, which are often not publicly available. In this work, we introduce *mr.mash-rss*, an extension of the *mr.mash* model that requires only summary statistics from Genome-Wide Association Studies (GWAS) and linkage disequilibrium (LD) estimates from a reference panel. By using summary data, we achieve the twin goal of increasing the applicability of the *mr.mash* model to data sets that are not publicly available and making it scalable to biobank-size data. Through simulations, we show that *mr.mash-rss* is competitive with, and often outperforms, current state-of-the-art methods for single- and multi-phenotype polygenic prediction in a variety of scenarios that differ in the pattern of effect sharing across phenotypes, the number of phenotypes, the number of causal variants, and the genomic heritability. We also present a real data analysis of 16 blood cell phenotypes in the UK Biobank, showing that *mr.mash-rss* achieves higher prediction accuracy than competing methods for the majority of traits, especially when the data set has smaller sample size.

## Author summary

Polygenic prediction refers to the use of an individual's genetic information (*i.e.*, genotypes) to predict traits (*i.e.*, phenotypes), which are often of medical relevance. It is known that some phenotypes are related and are affected by the same genotypes. When this is the case, it is possible to improve the accuracy of predictions by using methods that model multiple phenotypes jointly and account for shared effects. *mr.mash* is a recently developed multi-phenotype method that can learn which effects are shared and has been

analyses, and the compiled results generated from our simulations have been deposited on Zenodo (https://doi.org/10.5281/zenodo.14262333). The methods are implemented in the R package mr. mash.alpha, available for download at https:// github.com/stephenslab/mr.mash.alpha.

**Funding:** Research reported in this publication was supported by the National Institute of General Medical Sciences of the National Institutes of Health under Award Number R35GM146868 to FM. The content is solely the responsibility of the authors and does not necessarily represent the official views of the National Institutes of Health. PS acknowledges support from Open Discovery Innovation Network (ODIN) under grant number NNF20SA0061466. The funders had no role in study design, data collection and analysis, decision to publish, or preparation of the manuscript.

**Competing interests:** The authors have declared that no competing interests exist.

shown to improve prediction. However, *mr.mash* requires large data sets of genetic and phenotypic information collected at the individual level. Such data are often unavailable due to privacy concerns, or are difficult to work with due to the computational resources needed to analyze data of this size. Our work extends *mr.mash* to require only summary statistics from Genome-Wide Association Studies, which are usually publicly available, instead of individual-level data. In addition, the computations using summary statistics do not depend on sample size, making the newly developed *mr.mash-rss* scalable to extremely large data sets. Using simulations and real data analysis, we show that our method is competitive with other methods for polygenic prediction.

## Introduction

Predicting complex trait phenotypes from genotypes is a central task of a few branches of quantitative genetics. In agricultural breeding, there is interest in predicting breeding values (EBV) to select the best individuals for reproduction and achieve an increase in performance over generations [1]. In human genetics, predicting medically relevant phenotypes such as disease risk via polygenic scores (PGS) is important to stratify the population and identify individuals with greater genetic risk [2]. Finally, with the advent of transcriptome-wide association studies (TWAS), predicting gene expression as an intermediate step has become of interest [3]. In all these applications, accurate predictions are important. The response to artificial selection is directly proportional to the accuracy of EBVs [4]. Precise identification of individuals at risk for a particular disease requires accurate PGS [2]. The power to discover gene-phenotype associations in TWAS depends on the accuracy of gene expression prediction [5].

Technically, phenotypic prediction is achieved by modeling the phenotype of interest as a multiple regression on genotypes at a set of genetic variants [6]. Both frequentist and Bayesian approaches to multiple regression have been developed for and/or applied to this task, with accuracy spanning from very low to high depending on the genetic architecture of the trait analyzed [7–12]. Multiple phenotypes may be genetically correlated due to pleiotropy (*i.e.*, the sharing of causal variants across traits). In that case, modeling these phenotypes jointly via multivariate multiple regression methods can improve effect sizes estimates by leveraging effect sharing and, thus, increase prediction accuracy [13–17]. Integrative approaches that combine multiple single-phenotype PGSs across phenotypes have also been shown to improve prediction accuracy [18, 19].

Recently, Morgante *et al.* (2023) introduced the "Multiple Regression with Multivariate Adaptive Shrinkage" or "*mr.mash*" [20]. *mr.mash* is a Bayesian approach to multivariate multiple regression that is able learn complex patterns of effect sharing across phenotypes directly from the data. This is achieved through the use of flexible priors on the effect sizes across phenotypes and an empirical Bayes (EB) framework to adapt these priors to the data. Computational efficiency is achieved by using Variational Inference (VI) as opposed to the more expensive Markov Chain Monte Carlo (MCMC) methods. For a detailed account of VI and EB in this context, including the advantages, we direct the reader to [20]. Using multi-tissue gene expression prediction from cis-genotypes as an example, the authors showed that *mr.mash* is competitive in terms of both prediction accuracy and speed [20]. However, while powerful, *mr.mash* has some limitations. First, *mr.mash* requires individual-level data, *i.e.*, genotypes and phenotypes for each individual and, mainly for privacy reasons, these data are rarely publicly available [21]. Second, *mr.mash* does not scale well to datasets with very large sample size,

such as modern biobanks. These weaknesses limit the use of *mr.mash* for PGS prediction in human genetics.

In this work, we overcome both these limitations by introducing "*mr.mash* Regression with Summary Statistics" or "*mr.mash-rss*", an extension of *mr.mash* that only requires summary-level data. These are effect sizes and their standard errors (or Z-scores) from univariate Genome-Wide Association Studies (GWASs) and Linkage Disequilibrium (LD) estimates from reference panels, which are usually publicly available [21]. *mr.mash-rss* shares some features with the established Multivariate Adaptive Shrinkage (*mash*) [22], in that they both use the same mixture-of-multivariate-Normals prior on the effect sizes to leverage effect sharing across conditions (*e.g.*, different phenotypes), and the EB approach to adapt the prior to the data. In fact, both *mr.mash* and *mr.mash-rss* "borrow" this framework that was introduced with *mash*. However, while *mash* assumes that the input summary statistics come from independent variables (*i.e.*, it does not deal with LD), *mr.mash-rss* takes a full multivariate multiple regression approach and adjusts effect sizes for both sharing across conditions and correlations among variables (*i.e.*, it takes LD into account). We test *mr.mash-rss* in the task of PGS prediction for multiple phenotypes jointly via simulations in several scenarios and show that it is competitive in terms of prediction accuracy with currently available methods. We then confirm these results in the analysis of real data for 16 blood cell traits in the UK Biobank [23, 24].

## Description of the method

The multivariate multiple regression is used to model the effects of several predictor variables $X$ on multiple responses $Y$ jointly:

$$
\begin{aligned}
Y &= XB + E \\
E &\sim MN_{n \times r}(\mathbf{0}, I_n, V)
\end{aligned}
\tag{1}
$$

where $Y \in \mathbb{R}^{n \times r}$ is the response matrix for $r$ responses (phenotypes in our case) in $n$ individuals, $X \in \mathbb{R}^{n \times p}$ is the predictor matrix for $p$ predictors (genetic variants in our case) in $n$ individuals, $B \in \mathbb{R}^{p \times r}$ is the matrix of effects for $p$ predictors and $r$ responses, and $E \in \mathbb{R}^{n \times r}$, is the matrix of residuals for $r$ responses for $n$ individuals. The residuals follow a Matrix Normal distribution with mean $\mathbf{0}$ (an $n \times r$ matrix of zeroes), covariance across individuals $I_n$ (an $n \times n$ identity matrix), and covariance across responses $V$ (an $r \times r$ positive definite matrix).

*mr.mash* adopts a Bayesian approach by imposing a prior on the effects:

$$
b_j \mid w_0, \mathscr{S}_0 \sim \sum_{k=1}^{K} w_{0,k} N_r(\mathbf{0}, S_{0,k}), \quad j = 1, \ldots, p.
\tag{2}
$$

where $b_j$ is an $r$-vector that captures the effects of predictor $j$, and $b_j^\intercal$ is the $j^{th}$ row of $B$. Thus, the effects are assumed to be identically distributed as a mixture of $r$-variate Normals with $K$ components. The prior is determined by $w_0 := (w_{0,1}, \ldots, w_{0,K})$, the set of non-negative mixture weights, and $\mathscr{S}_0 := \{S_{0,1}, \ldots, S_{0,K}\}$, the set of $r \times r$ covariance matrices across responses. The elements of $\mathscr{S}_0$ are prespecified and are intended to capture plausible patterns of effect sharing across responses [20].

To make the model fit computationally efficient for large datasets, *mr.mash* approximates $p(B|X, Y, V, w_0, \mathscr{S}_0)$, the true posterior distribution of the regression coefficients, through variational inference, which uses optimization techniques to find the best approximation within a chosen family of distributions [25]. The optimal approximation is determined by maximizing the evidence lower bound (ELBO), a lower bound on the model's marginal likelihood. In addition, *mr.mash* also estimates $w_0$ (and $V$) from the data by maximizing the ELBO,

thereby adapting the prior to the data. This whole procedure has been termed variational empirical Bayes [26].

### Extension of *mr.mash* to summary statistics

Following the approach of [27], we express the updates in *mr.mash* in terms of sufficient statistics. The likelihood for the *mr.mash* model is

$$
\begin{aligned}
\ell(\boldsymbol{B}; \boldsymbol{X}, \boldsymbol{Y}, \boldsymbol{V}) \quad &:= \quad MN_{n \times r}(\boldsymbol{Y}; \boldsymbol{XB}, \boldsymbol{I}_n, \boldsymbol{V}) \\
&= \quad |2\pi\boldsymbol{V}|^{-n/2} \exp\{-\tfrac{1}{2}\mathrm{tr}[\boldsymbol{V}^{-1}(\boldsymbol{Y}^{\mathsf{T}}\boldsymbol{Y} - \boldsymbol{Y}^{\mathsf{T}}\boldsymbol{XB} - \boldsymbol{B}^{\mathsf{T}}\boldsymbol{X}^{\mathsf{T}}\boldsymbol{Y} + \boldsymbol{B}^{\mathsf{T}}\boldsymbol{X}^{\mathsf{T}}\boldsymbol{XB}]\}.
\end{aligned}
\tag{3}
$$

We can see that $\boldsymbol{X}^{\mathsf{T}}\boldsymbol{X}$, $\boldsymbol{X}^{\mathsf{T}}\boldsymbol{Y}$, and $\boldsymbol{Y}^{\mathsf{T}}\boldsymbol{Y}$ are sufficient statistics for the likelihood. Thus, the *mr.mash* model can be fitted using expressions based only on these sufficient statistics (see S1 Text for detailed derivations) to obtain the same results as using individual-level data $\boldsymbol{X}$ and $\boldsymbol{Y}$.

The sufficient statistics can be recovered from effect sizes and their standard errors (or Z-scores) from GWAS and LD estimates, following steps provided in [27] and S1 Text. We call *mr.mash* with summary statistics *mr.mash-rss*. However, it should be noted that while $\boldsymbol{X}^{\mathsf{T}}\boldsymbol{Y}$ can be recovered exactly, $\boldsymbol{X}^{\mathsf{T}}\boldsymbol{X}$ is only approximated when LD estimates come from reference panels, rather than from the data that generated GWAS summary statistics [27]. Thus, using summary data can be seen as fitting the *mr.mash* model using an approximation to the likelihood in 3 [27]. The quality of the approximation depends on how closely the LD reference panel matches the GWAS summary statistics. Quality control should therefore be performed on summary statistics and LD before model fitting [27, 28]. In addition, $\boldsymbol{Y}^{\mathsf{T}}\boldsymbol{Y}$ may not be available. However, this quantity is not strictly necessary, unless $\boldsymbol{V}$ is estimated within the *mr.mash-rss* algorithm [27]. While *mr.mash* has a way to deal with missing values in $\boldsymbol{Y}$, *mr.mash-rss* assumes the summary statistics be computed using the same individuals for each response (*i.e.*, there are no missing values in $\boldsymbol{Y}$).

The methods introduced in this paper are implemented in the R package [29] `mr.mash.alpha` which is available for download at https://github.com/stephenslab/mr.mash.alpha.

## Verification and comparison

### Simulations using UK Biobank genotypes

We devised a simulation study where the goal was to compare *mr.mash-rss* and other competing methods at computing PGS for multiple phenotypes from summary data. We used real genotypes from the UK Biobank array data for $n = 105,000$ nominally unrelated White British individuals that were randomly sampled. After applying a series of filters (see S1 Text for details), the data included $p = 595,071$ genetic variants.

We simulated $r = 5$ phenotypes according to three scenarios that differed in the structure of the effect sharing across phenotypes. Causal variants (5,000 for all scenarios) were randomly sampled from all genetic variants.

1. "Equal Effects", where each causal variant affects all the phenotypes and has the same effect across phenotypes. The per-phenotype proportion of variance explained by the causal variants or genomic heritability ($h_g^2$) is equal to 0.5.

2. "Mostly Null", where the causal variants affect only the first phenotype with $h_g^2$ equal to 0.5, while the remaining phenotypes are affected only by a non-genetic component (*i.e.*, $h_g^2 = 0$)

3. "Shared Effects in Subgroups", where the effect of each causal variant is drawn such that it is equally likely to be shared (but not be equal) in phenotypes 1 through 3 or to be shared (but not be equal) in phenotypes 4 and 5. The per-phenotype $h_g^2$ is 0.3 in phenotypes 1–3 and 0.5 in phenotypes 4 and 5.

These three scenarios were similar to those used in [20], but some parameters (*e.g.*, number of causal variants) were modified to reflect more closely the genetic architecture of complex traits, rather than gene expression. We also simulated a few scenarios based on the Equal Effects scenario (*i.e.*, equal effects of the causal variants across phenotypes) to assess the effect of genomic heritability, polygenicity (*i.e.*, number of causal variants), and number of phenotypes modeled on the performance of the methods:

4. "Low $h_g^2$", where the per-phenotype $h_g^2$ is 0.2.

5. "High Polygenicity", where the number of causal variants is 50,000.

6. "More Phenotypes", where the number of simulated phenotypes is 10.

For each of the scenarios above, we simulated 20 replicates. Per-phenotype prediction accuracy was computed as the $R^2$ from the linear regression of the true phenotypes on the predicted phenotypes for the test set individuals, which consisted of 5,000 randomly sampled individuals from the total of 105,000. This metric has the attractive property that its upper bound is $h_g^2$ [30].

## Methods compared

We compared *mr.mash-rss* to a few competing methods that satisfied the following requirements: (1) can be fitted with only summary data; (2) do not require a validation data set to tune model parameters; (3) for multivariate methods, are able to model at least 5 phenotypes *jointly*. This resulted in the choice of the following methods:

1. *LDpred2-auto*. This is a univariate Bayesian method that imposes a two-component mixture prior on the regression coefficients, consisting of a point-mass at 0 and a zero-centered Normal distribution [10]. This method is labelled "LDpred2" in the results.

2. *SBayesR*. This is a univariate Bayesian method that imposes a four-component mixture prior on the regression coefficients, consisting of a point-mass at 0 and three zero-centered Normal distributions, each with a different variance [9]. This method is labelled "SBayesR" in the results.

3. *SmvBayesC*. This is a multivariate Bayesian method that imposes a two-component mixture prior on the regression coefficients across phenotypes, consisting of a point-mass at 0 and a zero-centered multivariate Normal distribution [31, 32]. This method allows for each genetic variant to affect any combination of phenotypes. This method is labelled "SmvBayesC" in the results. We also tested a "restrictive" version that allows for each genetic variant to affect all or none of the phenotypes only [14, 31]. This method is labelled "SmvBayesC-rest" in the results.

We also included two 2-step approaches:

4. *MTAG+LDpred2-auto*. The first step uses MTAG, which is a multivariate method that adjusts univariate ordinary least squares (OLS) summary statistics based on the (estimated) correlation between the effects across phenotypes [33]. Because MTAG does not account for LD between variants, MTAG-adjusted summary statistics are then fed to

*LDpred2-auto* in the second step. This method is labelled "MTAG+LDpred2" in the results.

5. *wMT-SBLUP*. This is a method that uses SBLUP to convert univariate OLS summary statistics into univariate Best Linear Unbiased Predictor (BLUP) estimates in the first step. In the second step, the univariate BLUP estimates for multiple phenotypes are adjusted based on weights that take into account the genetic correlations among phenotypes and the sample size from which the summary statistics were computed [34]. This method is labelled "wMT-SBLUP" in the results.

Each method was fitted for each chromosome separately using summary statistics calculated using only the training set individuals. The summary statistics (*i.e.*, effect sizes and standard errors) were computed from univariate simple linear regression of each phenotype on each genetic variant, one at a time. Each phenotype was quantile normalized before the analysis. LD between each pair of variants was computed using 146,288 nominally unrelated White British individuals that did not overlap with the 105,000 individuals used for the rest of the analyses. Correlations between variants that were more than 3 cM apart were set to 0 to create a "banded" LD matrix [10]. We fitted *mr.mash-rss* including both "canonical" and "data-driven" covariance matrices (see S1 Text and [20] for details).

## Results

In the Equal Effects scenario (Fig 1A), the three fully multivariate methods (*i.e.*, *mr.mash-rss*, and the two versions of *SmvBayesC*) performed better than the two univariate methods. This is expected because the univariate methods assume independence among genetic effects across phenotypes and are unable to learn the pattern of equal genetic effects. Among the multivariate methods, *mr.mash-rss* produced higher accuracy than *SmvBayesC*. The "restrictive" version of *SmvBayesC* performed as well as the unrestricted one because this scenario meets one of the effect combinations allowed by this less flexible method. The two 2-step approaches (*i.e.*,

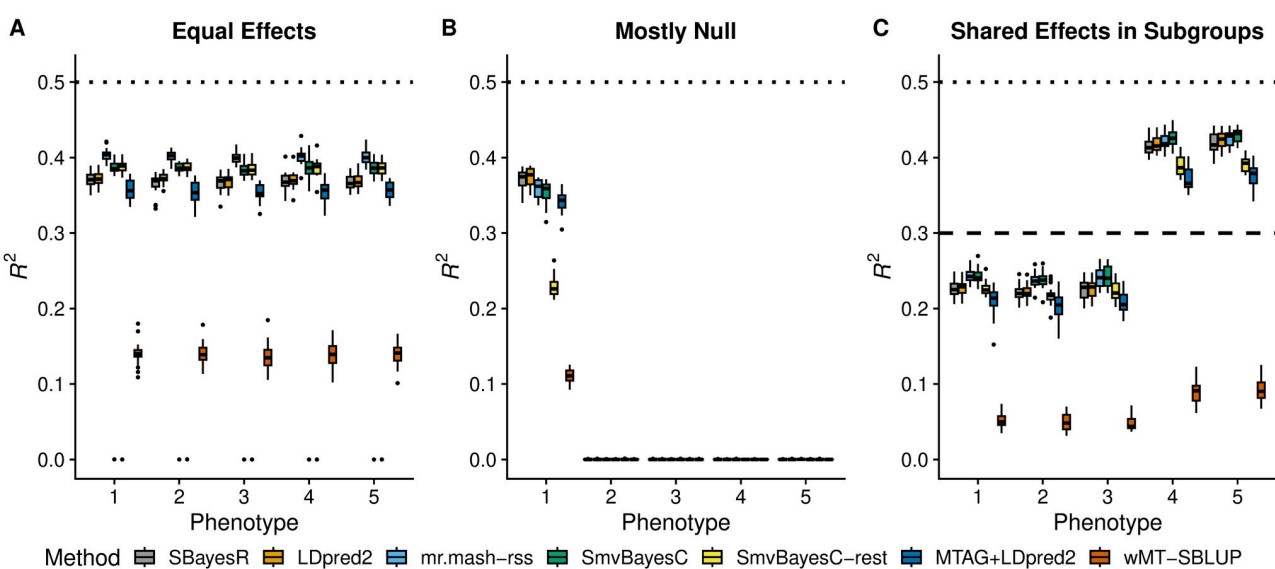

**Fig 1. Prediction accuracy in simulations with different patterns of effect sharing across phenotypes.** Each panel summarizes the accuracy of the test set predictions in 20 simulations. The thick, black line in each box gives the median $R^2$. The dotted and dashed lines give the maximum accuracy achievable, *i.e.*, the simulated $h_g^2$.

*MTAG+LDpred2-auto* and *wMT-SBLUP*) did not perform as well as the other methods. In particular, *wMT-SBLUP* performed substantially worse than the other methods. We attribute the poor performance to the infinitesimal architecture assumption that *wMT-SBLUP* makes that does not match our simulation scenarios.

In the Mostly Null scenario (Fig 1B), the genetic effects are present only in the first phenotype. Thus, joint modeling of all the phenotypes is not expected to produce any increase in accuracy compared to a phenotype-by-phenotype analysis. In phenotype 1, while *SBayesR* and *LDpred2-auto* were the most accurate methods, *mr.mash-rss* only had slightly lower mean $R^2$. As for *SmvBayesC*, the full version performed only slightly worse than *mr.mash-rss*; however, the "restrictive" version performed much worse. This observation is expected, given that the prior of *SmvBayesC* "restrictive" only allows for the effects to be present in all or none of the phenotypes. *MTAG+LDpred2-auto* did a little worse than the other multivariate methods, while *wMT-SBLUP* performed the worst.

In the Shared Effects in Subgroups scenario (Fig 1C), *SBayesR*, *LDpred2-auto*, *SmvBayesC*, and *mr.mash-rss* performed very similarly in phenotypes 4 and 5, with *SmvBayesC* having slightly higher accuracy than the other methods. On the other hand, *SmvBayesC* and *mr.mash-rss* outperformed the univariate methods in phenotypes 1–3. This can be explained by the slightly higher sharing of effects across phenotypes, the larger number of phenotypes with shared effects, and the lower $h_g^2$, which make the advantage of a multivariate analysis more clear than in phenotypes 4 and 5. The prior of *SmvBayesC* "restrictive" is not well-suited for this scenario, which resulted in this method not performing well across phenotypes. Similarly, MTAG's assumption that all genetic variants have the same effect sharing patterns across traits is clearly violated in this scenario. This resulted in *MTAG+LDpred2-auto* not performing as well as the other methods, with the exception of *wMT-SBLUP*, which again performed poorly.

In the Low $h_g^2$ scenario (Fig 2A), the three fully multivariate methods performed better than the two univariate methods. In addition, the relative improvement provided by the multivariate methods was larger than in the Equal Effects scenario with $h_g^2 = 0.5$ (Fig 1A). With smaller signal-to-noise ratio, it is harder to estimate effects accurately. Multivariate methods can

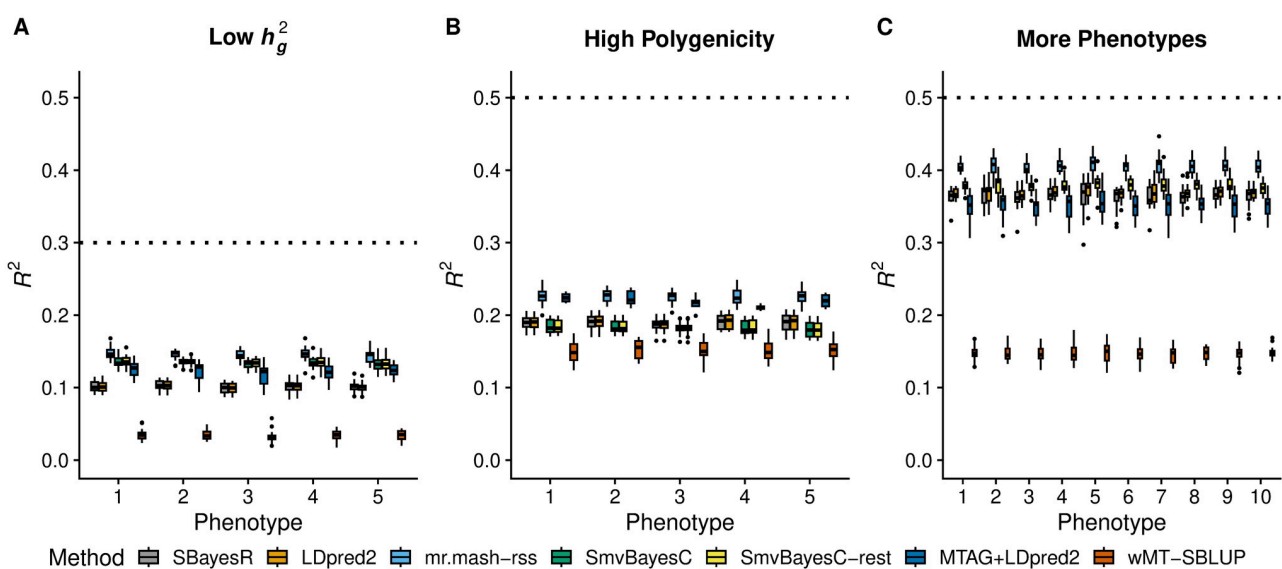

**Fig 2. Prediction accuracy in simulations with different genetic architecture.** Each panel summarizes the accuracy of the test set predictions in 20 simulations. The thick, black line in each box gives the median $R^2$. The dotted lines give the maximum accuracy achievable, *i.e.*, the simulated $h_g^2$.

borrow information across phenotypes and improve accuracy. In this scenario, *MTAG+-LDpred2-auto* also showed better performance than the univariate methods. *mr.mash-rss* was the best performing method, while *wMT-SBLUP* was the worst in this scenario.

In the High Polygenicity scenario (Fig 2B), prediction accuracy achieved by all the methods was much lower than in the Equal Effects scenario with 5,000 causal variants (Fig 1A). This is expected since each causal variant explains a much smaller proportion of phenotypic variance and, consequently, the effects are harder to estimate accurately. However, *mr.mash-rss* substantially outperformed both univariate and fully multivariate competing methods. *SmvBayesC* could not adapt well to this scenario, providing accuracies that are similar to or even lower than *SBayesR* and *LDpred2-auto*. *MTAG+LDpred2-auto* was very competitive with *mr.mash-rss*, essentially matching its performance for almost all phenotypes, while *wMT-SBLUP* was the worst in this scenario.

In the More Phenotypes scenario (Fig 2C), the results are very similar to the Equal Effects scenario with 5 phenotypes (Fig 1A). The relative improvement in accuracy provided by *mr.mash-rss* was, however, a little larger in this scenario because the method can borrow information across more phenotypes with equal effects. On the other hand, *SmvBayesC* "restrictive" did not benefit from the larger number of phenotypes and provided a relative improvement over the univariate methods that was similar to that of the Equal Effects scenario with 5 phenotypes. We could not run the full version of *SmvBayesC* in this scenario because it was too computationally intensive.

We note that *LDpred2-auto* had convergence issues when using MTAG-adjusted summary statistics, presumably due to it being designed to use OLS summary statistics. Thus, some trait-scenario combinations are based on fewer than 20 replicates for *MTAG+LDpred2-auto*.

To evaluate runtime, we selected chromosome 10 as a medium size chromosome and *LDpred2-auto*, *SmvBayesC*, and *mr.mash-rss* as the best performing methods in Shared Effects in Subgroups, *i.e.*, the most complex scenario in our simulations. The results confirm that *mr.mash-rss* is computationally efficient compared to the other multivariate method (S1 Table).

## Robustness to model misspecification

When analyzing real data, it is common to use LD matrices that are computed using a reference panel, rather than the individuals from whom the summary statistics were computed. It is well-known that issues with analyses with summary statistics arise when the reference panel and the GWAS population do not match [27, 35, 36]. Thus, it is important to assess the robustness of *mr.mash-rss* to the choice of the LD matrix. While we note that all the simulations above used an LD matrix that was computed using a subset of UK Biobank individuals that did not overlap with those used to compute the summary statistics, we also tested the performance of *mr.mash-rss* with truly external LD matrices. To do so, we used the same setting as the Equal Effects scenario, but computed LD matrices using 503 unrelated European individuals from the 1000G project [37]. This is a very small sample size that has been shown to be problematic when analyzing large GWAS samples [35]. Some preliminary testing highlighted that "banded" LD matrices (as used for the other simulations) resulted in convergence issues for the methods, while "block-diagonal" LD matrices computed as in [38] produced a better performance. Thus, we used the latter for the "External LD" simulation scenario. The results of this analysis are summarized in S1 Fig and show that, as expected, all the methods performed worse compared to the Equal Effects scenario. However, *mr.mash-rss* remained the best performing method overall.

*mr.mash-rss* assumes complete sample overlap across phenotypes. However, this might not be the case when analyzing real data. To investigate the robustness of *mr.mash-rss* to the violation of this assumption, we used the same setting as the Equal Effects scenario and assigned

missing values completely at random (MCAR) to individuals in the training set. This was done such that each individual had missing values in any combination of the five phenotypes with equal probability. We simulated two scenarios, one where 20% of the individuals had missing phenotypes and another one where 80% of the individuals had missing phenotypes. The results, summarized in S2 Fig, show that *mr.mash-rss'* performance was unchanged in the scenario with fewer missing phenotypes. In the scenario with a larger percentage of missing phenotypes, *mr.mash-rss'* prediction accuracy was now lower than *SmvBayesC*, but still higher than the univariate methods and similar to *MTAG+LDpred2-auto*.

In summary, these analyses show that *mr.mash-rss* is fairly robust to some model misspecifications.

## Applications

### Case study: Predicting blood cell traits in the UK Biobank

To evaluate *mr.mash-rss* on a real data application, we sought to predict blood cell traits from genotypes using the UK Biobank data. The UK Biobank is a dataset of roughly 500,000 individuals with genetic and phenotypic data [39]. We focused on a subset of 16 blood cell traits that have been used for quantitative genetic analyses in previous work [24]. After a series of filters (see S1 Text for details), our data consisted of $n$ = 244,049 individuals and $p$ = 1,054,330 Hap-Map3 variants, as has been previously recommended [40]. The 244K individuals were split into 5 non-overlapping groups to perform 5-fold cross-validation. Each method was trained on the data from 4 groups and prediction accuracy was computed in the remaining fifth group. This procedure was repeated five times, once for each fold. Given that *SmvBayesC* is too computationally intensive for this many phenotypes, *LDpred2-auto* suffered from convergence issues when using MATG-adjusted summary statistics, and *wMT-SBLUP* performed poorly in all simulation scenarios and does not account for sample overlap, we compared *mr. mash-rss*, *LDpred2-auto*, and *SBayesR* in the real data application.

The results of this analysis are summarized in Fig 3 and S2 Table. Overall, the three methods performed similarly. This result is similar to what we found in the "Shared Effects in Subgroup" simulation scenario, which was designed to be reflective of the complex genetic architecture and effect sharing patterns of actual complex traits. However, *mr.mash-rss* was the most accurate for 14 out 16 blood cell phenotypes. The relative change in mean prediction accuracy compared to *LDpred2-auto* ranged from -0.6% (Eosinophil Percentage) to 32.8% (Basophill Percentage), with an average of 5.4% (Table 1). The relative change in mean prediction accuracy compared to *SBayesR* ranged from -1.9% (Eosinophil Percentage) to 13.9% (Basophill Percentage), with an average of 2.7%(Table 1). The better performance of *SBayesR* compared to *LDpred2-auto* may be due to a more flexible prior that can better approximate the actual distribution of the genetic effects.

In accordance with the simulations, the improvement in accuracy tended to be largest for phenotypes with lower genomic heritability (though this relationship is only suggestive) as shown in Fig 4. With lower signal-to-noise ratio, leveraging the sharing of effects in a multivariate analysis can give greater improvements. This can be seen, for example, for Neutrophil Percentage ($h_g^2 = 0.16$; S3 Table), which has been shown to share putative causal variants with Lymphocyte Percentage (Fig. 3C in [24]) and is one of the phenotypes showing a greater improvement from using *mr.mash-rss*. On the other hand, the three platelet phenotypes have higher genomic heritability ($h_g^2 = 0.24$-$0.31$; S3 Table) and despite some sharing of causal variants (Fig. 3C in [24]), the improvements in accuracy from using *mr.mash-rss* are very small.

Previous analyses have shown that phenotypes with smaller sample size gain more advantage from multivariate modeling [20, 41]. We hypothesized that more substantial

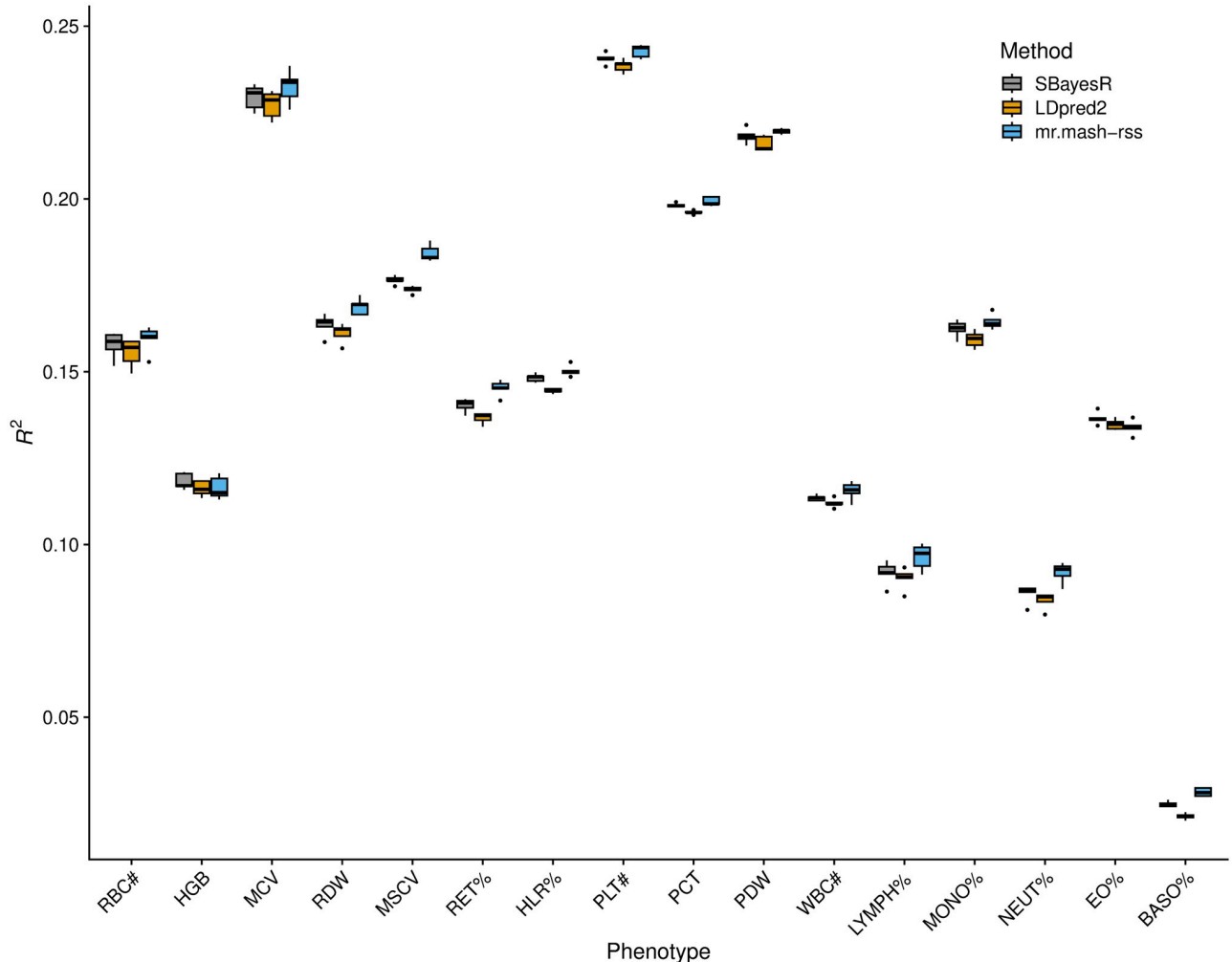

**Fig 3. Prediction accuracy for the 16 blood cell traits in the full UK Biobank data.** The thick, black line in each box gives the median $R^2$.

improvements in prediction accuracy from using *mr.mash-rss* could be obtained with a smaller sample size. Thus, we repeated the same analysis on 75,000 individuals, randomly sampled from the total of 244,049. The results, summarized in Fig 5 and S2 Table, showed that this is indeed the case.

In fact, the relative change in mean prediction accuracy compared to *LDpred2-auto* ranged from 7.2% (Monocyte Percentage) to 122.3% (Basophill Percentage), with an average of 23.3% (Table 1). This is about 4 times larger than the average relative change in mean prediction accuracy using the full data. The relative change in mean prediction accuracy compared to *SBayesR* ranged from -2.4% (Eosinophil Percentage) to 13.5% (Neutrophil Percentage), with an average of 5.2% (Table 1). This is about 2 times larger than the average relative change in mean prediction accuracy using the full data.

## Case study: Predicting more polygenic traits in the UK Biobank

We then sought to predict eight more polygenic phenotypes. Based on [42], we chose a group of phenotypes that have high pairwise genetic correlations; namely, body mass index (BMI),

**Table 1. Percentage change in mean $R^2$ of *mr.mash-rss* relative to *LDpred2-auto* and *SBayesR* for the 16 blood cell traits in the full and sampled UK Biobank data.**

| Phenotype | Full data | | Sampled data | |
|---|---|---|---|---|
| | *LDpred2-auto* | *SBayesR* | *LDpred2-auto* | *SBayesR* |
| Red Blood Cell Counts (RBC#) | 2.6 | 1.1 | 18.1 | 6.1 |
| Haemoglobin Concentration (HGB) | 0.1 | -1.6 | 13.5 | -0.7 |
| Mean Corpuscular Volume (MCV) | 2.3 | 1.3 | 7.7 | 1.3 |
| Red Blood Cell Volume Distribution Width (RDW) | 4.8 | 3.3 | 19.6 | 7.5 |
| Mean Sphered Cell Volume (MSCV) | 6.1 | 4.4 | 17.2 | 7.6 |
| Reticulocyte Percentage (RET%) | 6.3 | 3.5 | 20.1 | 8.3 |
| High Light Scatter Reticulocytes Percentage (HLR%) | 4.0 | 1.4 | 16.3 | 4.7 |
| Platelet Count (PLT#) | 1.8 | 0.9 | 11.3 | 4.3 |
| Plateletcrit (PCT) | 1.6 | 0.6 | 10.9 | 3.0 |
| Platelet Distribution Width (PDW) | 1.7 | 0.7 | 9.0 | 1.7 |
| White Blood Cell Count (WBC#) | 3.2 | 1.8 | 18.3 | 4.3 |
| Lymphocyte Percentage (LYMPH%) | 6.9 | 5.1 | 32.9 | 11.5 |
| Monocyte Percentage (MONO%) | 3.2 | 1.3 | 7.2 | -1.4 |
| Neutrophil Percentage (NEUT%) | 9.7 | 7.1 | 36.0 | 13.8 |
| Eosinophil Percentage (EO%) | -0.6 | -1.9 | 12.4 | -2.4 |
| Basophil Percentage (BASO%) | 32.8 | 13.9 | 122.3 | 13.5 |

Underlined are the negative values, *i.e.*, those instances where *mr.mash-rss* produces lower accuracy than the competing method.

trunk fat mass (TFM), body fat percentage (BFP), weight, waist circumference, hip circumference. We also chose two additional phenotypes, namely diastolic blood pressure (DP) and systolic blood pressure (SP), that are highly genetically correlated with each other and are moderately genetically correlated with the phenotypes in the first group. This was meant as a

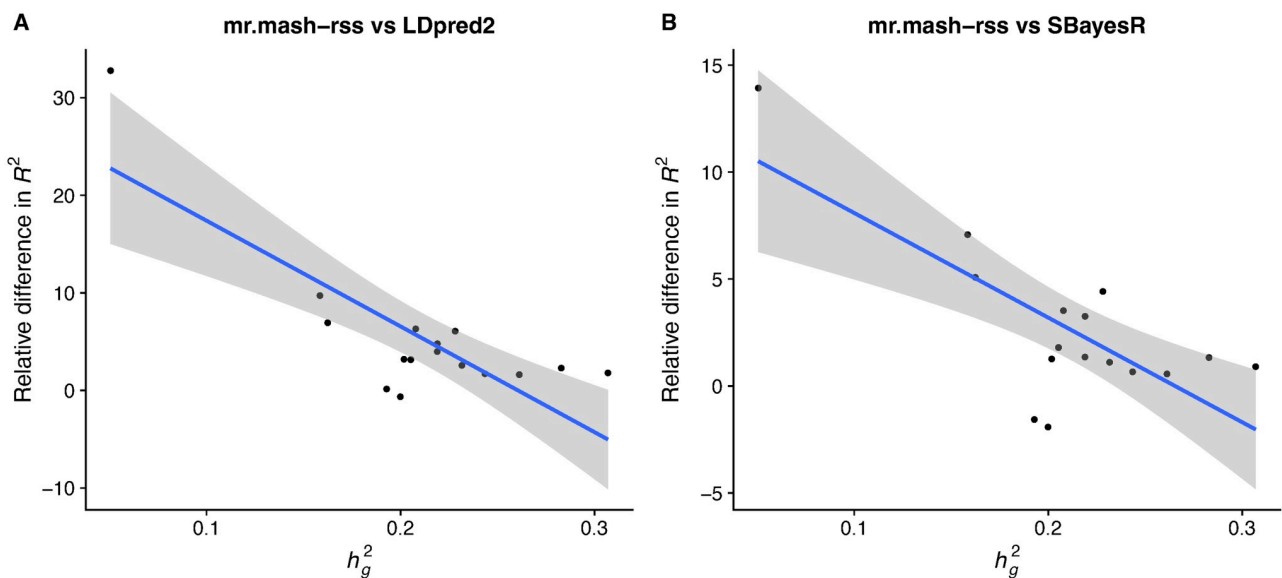

**Fig 4. Relationship between improvement in prediction accuracy and genomic heritability in the full UK Biobank data.** Phenotypes are plotted along the x-axis by their genomic heritability ($h_g^2$) and along the y-axis by the change in $R^2$ relative to the *LDpred2-auto* (Panel A) and *SBayesR* (Panel B); that is, ($R^2$(*mr.mash-rss*)—$R^2$(other method))/$R^2$(other method). The blue line represents the linear regression fit with 95% confidence bands.

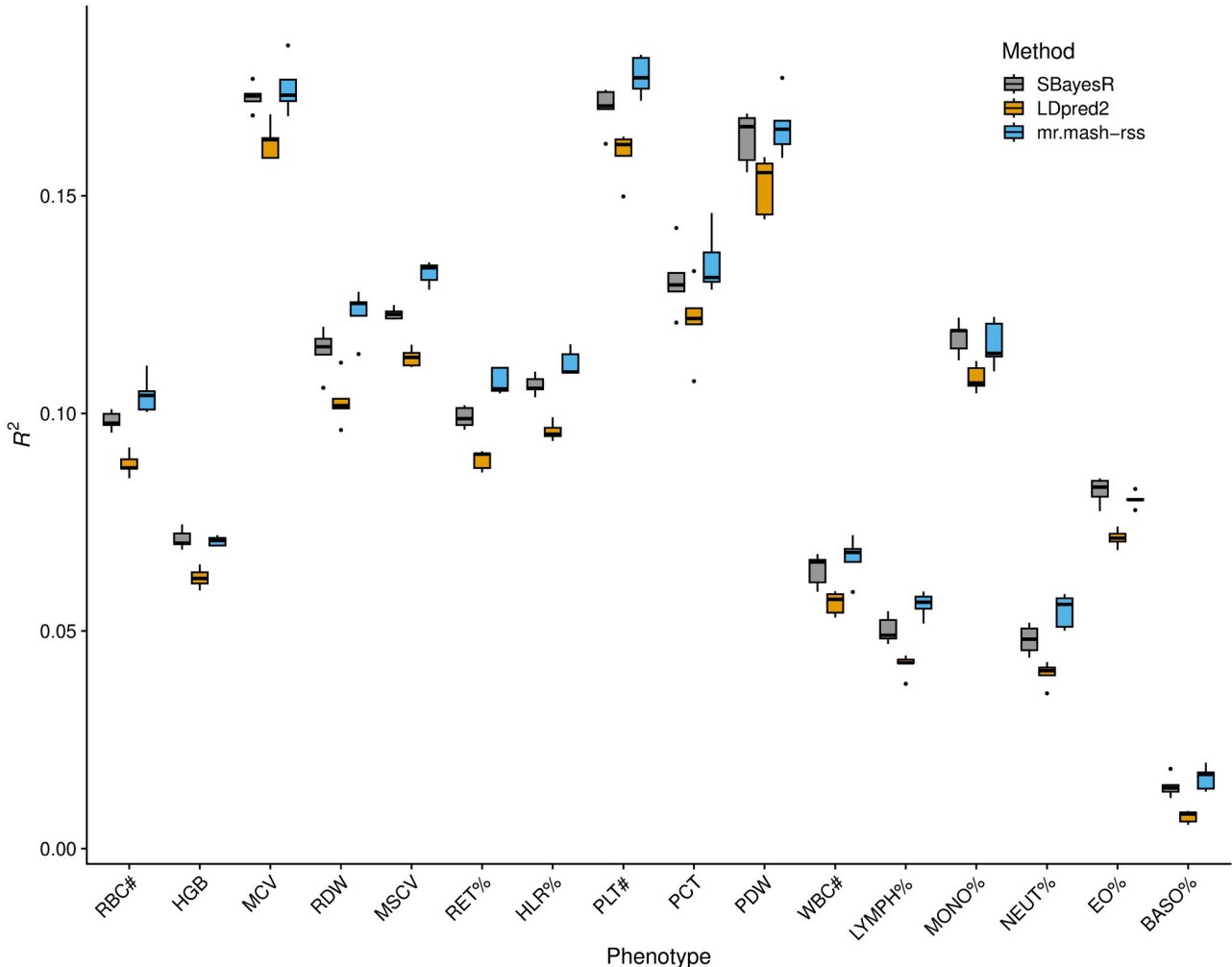

**Fig 5. Prediction accuracy for the 16 blood cell traits in the sampled UK Biobank data.** The thick, black line in each box gives the median $R^2$.

stress test of our methods, given that previous studies have shown that this type of variational empirical Bayes methods are usually less competitive with dense signals [12, 43].

The results of this analysis are summarized in S3 Fig and show that, as expected, *mr.mash-rss* was outperformed by the other methods for all the phenotypes in this case study. Previous studies have found the source of the under performance to be the M-step update for the prior mixture weights [12]. Here, anecdotally, we also observed the same phenomenon, whereby the weights on the null component and on the components with very small variance in the mixture tended to be over estimated, resulting in over shrinkage of the effects. [12] solved the issue by using a grid search and cross-validation approach to select the combination of mixture weights that maximizes prediction accuracy in a test set. However, this strategy is not feasible for *mr.mash-rss* wherein the number of mixture components is often more than 100.

Thus, we used a different strategy to try and improve *mr.mash-rss*' performance. In particular, we ran *mash* with the same mixture prior as *mr.mash-rss*, on the summary statistics for a subset of semi-independent LD-pruned genetic variants for all chromosomes. We extracted the estimated mixture weights, set the weight on the null component to 0.5, and rescaled the

other mixture weights accordingly (this step was necessary because *mash* underestimated the null weight due to it being run on a small sample of genetic variants). These mixture weights were fed to *mr.mash-rss*, which was then constrained to update the mixture weights for only the first 10 iterations. In this way, we maintained the adaptive nature of the empirical Bayes without incurring over shrinkage of the effects.

The results show that using this strategy (termed "mr.mash-rss mash" in the S3 Fig), the performance of *mr.mash-rss* improves for every phenotype, becoming comparable to the other methods' for DP, SP, hip, and weight, and superior for BMI.

## Discussion

In this work, we have introduced *mr.mash-rss*, the summary data version of a recently developed empirical Bayes multivariate multiple regression method [20]. Like *mr.mash*, *mr.mash-rss* enjoys (1) the ability to learn patterns of effect sharing across phenotypes; (2) the ability to model dozens of phenotypes jointly; (3) computational efficiency. Additionally, *mr.mash-rss* addresses two important limitations of *mr.mash* —the need for individual-level data and the lack of scalability to biobank-size data.

Through an array of simulations and real data analysis using the UK Biobank, we showed that *mr.mash-rss* is competitive with state-of-the-art univariate and multivariate PGS methods. Of note, *mr.mash-rss* outperformed competing methods in 14 out of 16 blood cell phenotypes, although the magnitude of the improvement varied across phenotypes, from modest to substantial. This highlights that the general *mr.mash* model can adapt to either more sparse (*e.g.*, for gene expression [20]) or more dense (*e.g.*, for complex traits) genetic architectures. We also showed that the improvement in prediction accuracy from using *mr.mash-rss* increased substantially with a smaller sample size. This holds good promise for improving prediction accuracy for phenotypes that are difficult to measure and in samples of individuals of non-European descent, which are usually much smaller [44]. In addition, the performance of the *mr.mash* model depends on the accuracy of the "data-driven" covariance matrices [20]. Thus, advances in covariance matrix estimation can potentially lead to improvements in prediction accuracy.

A limitation of *mr.mash-rss* is that it does not perform as well for very polygenic phenotypes as highlighted by the second data application. This is a known issue for this type of variational empirical Bayes methods [12, 43]. The problem stems from the update of the hyperparameters (the mixture weights, in particular) by maximization of the ELBO, which gets trapped in sub-optimal local optima [12, 45]. This issue can be overcome by using a grid search and cross-validation approach to select the combination of mixture weights that maximizes prediction accuracy in a test set [11, 12]. However, *mr.mash-rss* usually includes a large number of mixture components, which makes the grid search and cross-validation approach not feasible. We were able to ameliorate this problem by obtaining good initial estimates of the mixture weights with *mash*, which were then refined with a few iterations of *mr.mash-rss*. This simple strategy improved the prediction accuracy of *mr.mash-rss*, making it competitive for most traits analyzed. Future research is needed to find a more principled way to select hyperparameters that works well with arbitrary patterns of sparsity in the genetic architecture of complex traits.

Another limitation of *mr.mash-rss* is that it requires the summary statistics to be computed on the same samples for each phenotype. In other words, there should not be missing data in $Y$ in 1. Dealing with arbitrary patterns of missing data in multivariate models is not a trivial problem [46] and is an area where more research is needed. If individual-level data are available, missing values may be imputed before the prediction analysis. In fact, recent work has shown that imputing missing values results in improved prediction accuracy of PGS and

power in GWAS [47, 48]. Nonetheless, our results showed that *mr.mash-rss* is robust to a small to medium amount of missing phenotypes. In addition, in specific cases such as with complete sample non-overlap across phenotypes, some simplifications arise that allow for models like *mr.mash-rss* to be fitted efficiently [38].

Our work showed that *mr.mash-rss* is fairly robust to some forms of model misspecification (*i.e.*, external LD and sample non-overlap). However, model misspecification also arises with the use of "imperfect" summary statistics. For example, when summary statistics come from a meta-analysis of multiple cohorts, sample size is often different among genetic variants, and different biases and noise levels likely affect different cohorts [40]. One way to test the robustness of *mr.mash-rss* to different sources of model misspecification would be to use truly external summary statistics, possibly from a meta-analysis, and evaluate its performance in an independent cohort.

This work evaluated *mr.mash-rss* using continuous phenotypes. While the theory behind the method assumes the phenotypes to be continuous, it may be possible for *mr.mash-rss* to be applied to case-control phenotypes, in the same way as methods such *LDpred2-auto* and *SBayesR*, which also assume continuous phenotypes. An in-depth investigation of the performance of *mr.mash-rss* for case-control phenotypes is left for future work.

## Supporting information

**S1 Table. Summary statistics for runtime (in seconds) on chromosome 10 for the "Shared Effects in Subgroups" scenario.** For *LDpred2-auto*, the statistics are based on the sum of runtime across phenotypes. Each method was run using 4 CPUs.
(PDF)

**S2 Table. Mean prediction $R^2$ across test sets for the 16 blood cell traits in the full and sampled UK Biobank data.**
(PDF)

**S3 Table. Mean $h_g^2$ across training sets for the 16 blood cell traits in the full UK Biobank data.**
(PDF)

**S1 Fig. Prediction accuracy in simulations with external LD matrix.** The figure summarizes the accuracy of the test set predictions in 20 simulations of the Equal Effects scenario. The thick, black line in each box gives the median $R^2$. The dotted line gives the maximum accuracy achievable, *i.e.*, the simulated $h_g^2$.
(EPS)

**S2 Fig. Prediction accuracy in simulations with missing phenotypes.** Each panel summarizes the accuracy of the test set predictions in 20 simulations of the Equal Effects scenario. Panel A (B) includes the results of a scenario where 20% (80%) of the individuals have missing values in any combination of the 5 phenotypes. The thick, black line in each box gives the median $R^2$. The dotted line gives the maximum accuracy achievable, *i.e.*, the simulated $h_g^2$.
(EPS)

**S3 Fig. Prediction accuracy for 8 more polygenic traits in the full UK Biobank data.** The thick, black line in each box gives the median $R^2$.
(EPS)

**S1 Text. Detailed methods.** Detailed description of the methods, including: derivations of the *mr.mash-rss* algorithms; data preparation; simulations; methods compared; data analysis. (PDF)

## Acknowledgments

This research was conducted using the UK Biobank Resource under application number 129216. We thank Gao Wang, Yuxin Zou, Peter Carbonetto, and Matthew Stephens for useful discussions.

## Author Contributions

**Conceptualization:** Deborah Kunkel, Fabio Morgante.

**Data curation:** Fabio Morgante.

**Formal analysis:** Fabio Morgante.

**Funding acquisition:** Fabio Morgante.

**Investigation:** Deborah Kunkel, Fabio Morgante.

**Methodology:** Deborah Kunkel, Fabio Morgante.

**Project administration:** Fabio Morgante.

**Software:** Deborah Kunkel, Peter Sørensen, Vijay Shankar, Fabio Morgante.

**Supervision:** Fabio Morgante.

**Validation:** Deborah Kunkel, Fabio Morgante.

**Visualization:** Deborah Kunkel, Fabio Morgante.

**Writing – original draft:** Deborah Kunkel, Fabio Morgante.

**Writing – review & editing:** Deborah Kunkel, Peter Sørensen, Vijay Shankar, Fabio Morgante.

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
