## [Decision Letter · Decision Letter 0]

1 Jul 2024

Dear Dr Morgante,

Thank you very much for submitting your Methods entitled 'Improving polygenic prediction from summary data by learning patterns of effect sharing across multiple phenotypes.' to PLOS Genetics. Also, thank you for your patience with the review process; we apologize for the extensive delay in a decision. 

The manuscript was fully evaluated at the editorial level and by independent peer reviewers. The reviewers appreciated the attention to an important problem, but raised some substantial concerns about the current manuscript. While the manuscript is well written and presents valuable improvements over existing methods like mr.mash and mashR, it cannot be considered a significant advancement in its current form. A more comprehensive comparison with existing methods and an extended real data application are needed to ensure the robustness and applicability of the method in real scenarios. Based on the reviews, we will not be able to accept this version of the manuscript, but we would be willing to review a much-revised version. We cannot, of course, promise publication at that time.

If you decide to revise the manuscript for further consideration at PLOS Genetics, please aim to resubmit within the next 60 days, unless it will take extra time to address the concerns of the reviewers, in which case we would appreciate an expected resubmission date by email to plosgenetics@plos.org.

We are sorry that we cannot be more positive about your manuscript at this stage. Please do not hesitate to contact us if you have any concerns or questions.

Yours sincerely,

Anke Huels

Guest Editor

PLOS Genetics

Michael Epstein

Section Editor

PLOS Genetics

Reviewer's Responses to Questions

**Comments to the Authors:**

Reviewer #1: Thank you for the opportunity to review your manuscript titled "Improving polygenic prediction from summary data by learning patterns of effect sharing across multiple phenotypes." Your work on extending the mr.mash model to utilize summary statistics (mr.mash-rss) is a valuable contribution to the field of polygenic prediction. I have a few suggestions and comments that I hope you find helpful in refining your manuscript.

1. Similarity to Existing Methods:

* Your proposed method, mr.mash-rss, shows significant similarities to existing approaches, particularly mr.mash and mashR. While extending mr.mash to use summary statistics is a valuable adaptation, it would be beneficial to clearly articulate how your work distinguishes itself from these established methods. Highlighting the unique contributions and specific advancements of mr.mash-rss will help emphasize its novelty and impact.

2. Use of Variational Inference and Empirical Bayes:

* The incorporation of variational inference and empirical Bayes in mr.mash-rss is a noteworthy addition. However, the manuscript could benefit from a more detailed explanation of these techniques and their specific advantages in your context. Clarifying how variational inference improves computational efficiency and how empirical Bayes enhances adaptability will provide a clearer understanding of the strengths of your approach.

3. Technical Details and Assumptions:

* The manuscript mentions that while XTYXTY can be recovered exactly, XTXXTX is approximated using LD reference panels. It would be helpful to discuss the potential biases introduced by this approximation, especially when the reference panel does not perfectly match the study population. Additionally, addressing the assumption that summary statistics are computed using the same individuals for each response will clarify the applicability of your method in real-world scenarios where missing data are common.

4. Comprehensive Comparisons:

*Including comparisons with a wider array of existing methods, such as different Bayesian approaches and multi-trait models, will provide a clearer context for the improvements claimed by mr.mash-rss. Evaluating your method across various datasets with different characteristics (e.g., sample sizes, ethnic backgrounds, trait architectures) will strengthen the evidence for its general applicability and robustness.

5. Clarifying Contributions and Novelty:

* Clearly articulating the unique contributions of mr.mash-rss and how they advance the field will enhance the manuscript. Detailed explanations of the innovations in model design and the practical impact of these methodological improvements will help readers appreciate the significance of your work.

6. Consideration of Simpler Approaches:

* It might be valuable to explore whether a simpler approach using mashR could be effective. For instance, one could analytically compute summary statistics and then apply shrinkage using the LD matrix. This could offer a more straightforward and computationally efficient solution while still leveraging the benefits of LD information.

7. Constructive Suggestions:

* Provide more detailed insights into the implementation and advantages of variational inference in mr.mash-rss.

* Discuss the empirical Bayes techniques used and compare them with other adaptive methods.

* Acknowledge and discuss the limitations introduced by the approximation of XTXXTX and the assumptions made, particularly in diverse population contexts.

* Include a wider range of benchmarks to contextualize the improvements and demonstrate the practical significance of mr.mash-rss.

I hope these suggestions are helpful as you continue to refine your manuscript. Your work contributes valuable insights to the field of polygenic prediction, and I look forward to seeing how it progresses.

Reviewer #2: The authors have demonstrated that mr.mash-rss is a novel to learn pattern of shared effects across multiple phenotypes using GWAS summary statistics. The manuscript was well-written and clearly showed the purpose of the paper. While there is a major advancement in this work, there are several questions I would like to ask:

1. In simulations, I wonder if the authors could simulate using the same HapMap3 variants as SBayesR and LDpred2 were designed with this set of SNPs

2. On line 271-273, the author mentioned about the application to non-European descent, I would think this transferability is also due to the assumption of shared causal SNPs between ancestral groups.

3. I wonder how sensitive the method to GWASes with fewer overlapping samples and out-of-sample LD matrices (e.g. LD from 1000 Genomes).

4. I wonder if the author can evaluate the method in an independent cohort to demonstrate the power of the methods.

Reviewer #3: Kunkel et al. is a very nice paper that I really enjoyed reading. The method is concisely and clearly described, and the work is well organized. The paper extends the mr.mash method so that it can be fitted using only GWAS summary statistics, which if works, makes it much more useful in practice (as the authors note). Overall it’s a very nice study, however I believe it has some potential limitations that could be addressed. Below I provide some suggestions aimed at improving the work, where the main concern is robustness of the method in real scenarios and applicability. Also, I think the benchmarks could be extended to consider other potentially more effective multivariate methods.

Comments:

1. It seems to me that the method assumes that the covariance of genetic effects is known across outcomes (S_0). Do you estimate this using LD score regression? Also, it seems that the method also assumes Y’Y is known. However, this may not be directly estimable in practice, but could be obtained using LD score regression. LD score does however make different assumptions. I would be interested in seeing simulations and real data analyses that examined the performance when the GWAS summary stats are obtained on different cohorts, using partially and non-overlapping samples. Such analyses would in my mind represent the most useful applications scenarios for the method.

2. SBayesR and LDpred1/2, and other iterative methods do not always converge in practice when the summary statistics are somehow imperfect. This is not only an LD reference panel issue, which is obviously important as you note. E.g., in meta-analyses sample sizes often vary between SNPs, as not all SNPs may be called or imputed accurately across all cohorts. Also, different cohorts in meta-analyses may be subject to different biases and noise levels. Some of these biases could be examined in simulations, e.g. varying sample sizes, and LD reference quality. However, the best test is using real external GWAS summary statistics. You can examine public single-cohort summary statistics such as FinnGen or decode genetics summary statistics and predict into UKB.

3. The paper doesn’t examine applications to case-control outcomes. The methods that it compares against are generally all applicable to such outcomes (although the math involved generally assumes that they are quantitative). I see no clear reasons why mr.mash couldn’t be applied to such outcomes, and please explain if that’s not the case. Many disease outcomes are highly correlated, e.g. psychiatric disorders, cardiometabolic disorders, etc.

3.b. The real data considered all have relatively simple genetic architectures. I would also consider outcomes such as height, which could also be correlated to other anthropometric outcomes in the UKB. Or if possible, disease outcomes.

4. There are a couple of multivariate methods that I would recommend also considering. First, there’s SBLUP (Maier et al., Nat Comm 2018). I also recommend comparing with some of the following MTAG (Turley et al,. Nat Genet 2018), GSEM (Grotzinger et al., Nat Hum Behav 2019), or even multi-PGS (Albiñana et al. Nat Comm 2023). I do however concede that using some of these might be more complicated in practice as these are two-step approaches, in that you first have to generate multi-trait summary statistics and then fit using a single outcome PGS. However, I suspect that this could be more powerful in some settings, more generalizable and more robust. (I don’t expect you to compare against all of these, but I recommend some.)

5. There is no run time comparison, but this would be highly useful for people when deciding whether to use the software if this was presented.

6. It wasn’t clear to me exactly how the causal effects were sampled in the simulations. Did you sample their effects using equation 2? If so, what weights and values K did you use? Perhaps trying a couple combinations of these values would be of interest, to see how the performance varies with these.

**Have all data underlying the figures and results presented in the manuscript been provided?**

Reviewer #1: Yes

Reviewer #2: Yes

Reviewer #3: Yes

PLOS authors have the option to publish the peer review history of their article (what does this mean?). If published, this will include your full peer review and any attached files.

Reviewer #1: No

Reviewer #2: No

Reviewer #3: **Yes: **Bjarni Vilhjalmsson

---

## [Decision Letter · Decision Letter 1]

19 Oct 2024

Dear Dr Morgante,

Thank you very much for submitting your Methods entitled 'Improving polygenic prediction from summary data by learning patterns of effect sharing across multiple phenotypes.' to PLOS Genetics.

The manuscript was fully evaluated at the editorial level and by independent peer reviewers. The reviewers were generally satisfied with the revision but had a few minor comments that we ask you to address in a revised manuscript.

We therefore ask you to modify the manuscript according to the review recommendations. Your revisions should address the specific points made by each reviewer.

To resubmit, log into your Editorial Manager account and select the option 'Revise Submission' in the 'Submissions Needing Revision' folder.

Yours sincerely,

Anke Huels

Guest Editor

PLOS Genetics

Michael Epstein

Section Editor

PLOS Genetics

Reviewer's Responses to Questions

**Comments to the Authors:**

Reviewer #1: All comments have been addressed.

Reviewer #2: For integrative PRS, I would cite Albinana et al. Nat Comms, 2024 (https://www.nature.com/articles/s41467-023-40330-w) and Truong et al. Cell Genomics, 2024 (https://www.cell.com/cell-genomics/fulltext/S2666-979X(24)00065-X) because they recently proposed complementary methods to perform PRS combination.

The authors nicely addressed all of my comments!

Reviewer #3: Thank you for addressing many of my concerns and the comments made by other reviewers. I believe the manuscript has improved substantially, although I still have a couple of perhaps minor comments.

On your response to comment 2, I still believe this would be the most fair comparison for a summary statistics based method, but perhaps you can highlight this in the discussion.

On your response to comment 3, I don’t understand your reasoning for not applying mr.mash-rss to binary outcomes, especially because you also note that the methods that you compare against do. I suspect that these outcomes would be more polygenic than molecular measurements, and would therefore look like the ones in Supplementary Figure 3. Interestingly, the results in Supplementary Figure 3 suggest that mr.mash-rss underperforms on highly polygenic traits like BMI. It’s ok if mr.mash-rss doesn’t outperform other methods in every way, and I think exploring a few common case-control outcomes would be of interest.

On comment 4, why didn’t compare against MTAG+SBayesR instead of MTAG+LDpred2, as SBayesR seemed to outperform LDpred2?

**Have all data underlying the figures and results presented in the manuscript been provided?**

Reviewer #1: Yes

Reviewer #2: Yes

Reviewer #3: Yes

PLOS authors have the option to publish the peer review history of their article (what does this mean?). If published, this will include your full peer review and any attached files.

Reviewer #1: No

Reviewer #2: No

Reviewer #3: No

---

## [Editor Report · Decision Letter 2]

27 Nov 2024

Dear Dr Morgante,

We are pleased to inform you that your manuscript entitled "Improving polygenic prediction from summary data by learning patterns of effect sharing across multiple phenotypes." has been editorially accepted for publication in PLOS Genetics. Congratulations!

Yours sincerely,

Anke Huels

Guest Editor

PLOS Genetics

Michael Epstein

Section Editor

PLOS Genetics

Aimée Dudley

Editor-in-Chief

PLOS Genetics

Anne Goriely

Editor-in-Chief

PLOS Genetics

Comments from the reviewers (if applicable):

**Data Deposition**

http://datadryad.org/submit?journalID=pgenetics&manu=PGENETICS-D-24-00499R2

**Press Queries**

---

## [Editor Report · Acceptance letter]

27 Dec 2024

PGENETICS-D-24-00499R2 

Improving polygenic prediction from summary data by learning patterns of effect sharing across multiple phenotypes 

Dear Dr Morgante, 

We are pleased to inform you that your manuscript entitled "Improving polygenic prediction from summary data by learning patterns of effect sharing across multiple phenotypes" has been formally accepted for publication in PLOS Genetics! Your manuscript is now with our production department and you will be notified of the publication date in due course.

With kind regards,

Livia Horvath

PLOS Genetics

On behalf of:
